# Oral Intake of the Commercial Probiotic Blend Synbio^®^ for the Management of Vaginal Dysbiosis

**DOI:** 10.3390/jcm12010027

**Published:** 2022-12-20

**Authors:** Alessandra Pino, Agnese Maria Chiara Rapisarda, Amanda Vaccalluzzo, Rosamaria Roberta Sanfilippo, Maria Magdalena Coman, Raffaela Luisa Grimaldi, Cinzia Caggia, Cinzia Lucia Randazzo, Nunziatina Russo, Marco Marzio Panella, Antonio Cianci, Maria Cristina Verdenelli

**Affiliations:** 1Department of Agricultural, Food and Environment, University of Catania, Santa Sofia Street 100, 95123 Catania, Italy; 2ProBioEtna S.r.l., Spin Off of the University of Catania, Santa Sofia Street 100, 95123 Catania, Italy; 3CERNUT, Interdepartmental Research Centre in Nutraceuticals and Health Products, University of Catania, A. Doria Street 6, 95125 Catania, Italy; 4Department of General Surgery and Medical Surgical Specialties, University of Catania, 95123 Catania, Italy; 5Synbiotec S.r.l., Gentile III da Varano Street, 62032 Camerino, Italy

**Keywords:** probiotic, vaginal microbiota, microbiome, vaginal dysbiosis

## Abstract

A healthy vaginal microbiota is Lactobacillus-dominated. Several factors can interfere with the state of balance leading to dysbiosis, such as vaginal infections caused by bacteria and Candida species. The present single-arm, uncontrolled open-label study aimed to evaluate the ability of the SYNBIO^®^ probiotic combination, taken as an oral formulation, to contribute to vaginal health. Thirty pre-menopausal participants were included in the study. Participants were instructed for daily oral intake of SYNBIO^®^ probiotic capsules for 15 days. Vaginal swabs were collected at baseline (T0), 15 days after the start of the treatment (T1), and 7 days after the end of the treatment (T2). Amsel criteria, Nugent score, and vaginal pH were evaluated at each sampling time. In addition, the participants’ quality of life was assessed by the WHOQOL-BREF questionnaire. The administration of SYNBIO^®^ once daily for 15 days resulted in a substantial improvement in the vaginal flora in terms of an increase in lactobacilli and a decrease in enterococci, staphylococci, *Gardnerella* spp., and *Candida* spp. According to the results, statistically significant changes in leucorrhoea, itching, and vulvo-vaginal erythema/edema as well as a decrease in all the Amsel criteria were recorded. The oral consumption of SYNBIO^®^ demonstrated enhanced benefits for vaginal health.

## 1. Introduction

It is now widely recognized that the vaginal microbiota (VM) is a dynamic balanced ecosystem consisting of bacteria, viruses, archaea, fungi, and protozoa [1,2]. Under physiological conditions, the VM plays a pivotal role in women’s health and reproduction, acting as a frontline defender against pathogenic microorganisms. As first reported by Donderlein [3], it is widely accepted that in healthy women of reproductive age, a balanced VM is dominated by lactobacilli, generally represented by a single species [4]. Lactobacilli, by producing metabolites (e.g., lactic acid), antimicrobial substances such as bacteriocins and hydrogen peroxide, and interacting with the host’s innate immune system, can protect the vaginal mucosa from non-indigenous and potentially harmful microorganisms [5]. The vaginal ecosystem in women is highly dynamic in terms of microbial species due to several endogenous and exogenous factors that eventually affect vaginal homeostasis. An abnormal microbial composition in the vagina causes an increased growth of opportunistic pathogens, contributing to the onset of disease-related signs and symptoms [6]. In detail, in the case of non-Lactobacillus-dominant VM, a high risk for adverse health outcomes is reported; a reduction in or loss of lactobacilli is associated with common vaginal infections, such as bacterial vaginosis (BV) and vulvovaginal candidiasis (VVC), sexually transmitted infection (STI) [7], and spontaneous preterm birth [8,9]. Bacterial vaginosis is frequently associated with an increased abundance of facultative and anaerobic microorganisms, which are responsible for odorous vaginal discharge, burning during urination, and itching [10]. The prevalence of BV includes between 23% and 29% of women in reproductive age [11]. Concerning VVC, a common vaginal mucosa infection caused by Candida species, vulvar pruritus and burning, as well as vaginal dyspareunia and dysuria, are the main clinical symptoms. It is estimated that more than 75% of women experience at least once *Candida* infection in their lifetime and that recurrent VVC is frequent [12,13]. This is why preserving a balance in the vaginal microbial composition is essential to guarantee an adequate host–microbial interaction that ensures a healthy vaginal microbiota. First-line treatment for BV includes the use of topical or systemic antibiotics, such as metronidazole and clindamycin, whereas polyenes (e.g., nystatin) and azoles are usually administrated in the case of VVC [14]. Unfortunately, antibiotics and antifungal drugs are not always effective, since the microorganisms responsible for BV and VVC can resist these treatments by forming biofilms or by acquiring resistance [15,16]. In addition, antibiotics do not act selectively on the causative agent of the infection and negatively affect endogenous lactobacilli, in turn promoting BV and potentially VVC recurrence [17]. Based on this evidence, alternative solutions to balance the VM and maintain a healthy vaginal ecosystem are needed. Recent research on probiotics, “live microorganisms that, when administered in adequate amounts, confer a health benefit on the host” [18], has highlighted the ability of some probiotic strains or their combinations to reestablish the balance of the vaginal homeostasis, counteracting the onset of infections or the reappearance of relapses [19,20]. As reported by various meta-analyses and systematic reviews, probiotic bacteria can be considered a promising alternative to antibiotics in both the treatment of vaginal infections and the maintenance of healthy VM [13,20]. In fact, probiotics may confer several beneficial effects on the female genital tract through different mechanisms, such as the production of lactic acid and hydrogen peroxide to restore a low vaginal pH, adhesion to vaginal walls in order to inhibit the attachment of pathogenic bacteria, and the use of their same nutrients to decrease their growth [21,22,23].

The commercial probiotic blend SYNBIO^®^ is a combination of *Lacticaseibacillus rhamnosus* IMC 501^®^ and *Lacticaseibacillus paracasei* IMC 502^®^ strains formulated by SYNBIOTEC Srl (Camerino, Italy). The aforementioned strains were previously characterized as showing in vitro inhibitory activity against pathogenic bacteria and yeast [24]; an ability to adhere to HT-29 and HeLa cells [21,25]; and in vitro adhesion to vaginal epithelium removing or inhibiting the adhesion of Candida species [21,22]. The administration of SYNBIO^®^ as vaginal suppositories (Synbio^®^ gin) proved to be suitable in restoring and maintaining a normal vaginal microbiota in apparently healthy women [26].

The present study aimed to evaluate the in vivo effectiveness of orally administrated SYNBIO^®^ to treat vaginal dysbiosis. To achieve this goal, clinical and microbiological parameters were evaluated.

## 2. Materials and Methods

### 2.1. Study Design and Study Population

A single-arm, uncontrolled open-label trial was conducted between January 2022 and May 2022 at the Gynecological Service of the Department of General Surgery and Medical Surgical Specialties, Gynecological Clinic, University of Catania (Catania, Italy). The trial involved thirty (30) pre-menopausal participants, aged between 18 and 45 years. The inclusion criteria consisted of: the presence of vaginal sign or symptom (leucorrhoea, burning, itching, and subjective vaginal discomfort), a Nugent score ≥ 7 [27], the presence of at least 3 Amsel criteria [10], a lactobacillary grade ≥ 2 (LBG) (according to Donders classification [28]), and the presence of blastospores and/or pseudohyphae evaluated by microscopy. Exclusion criteria were: the presence of severe vulvovaginal symptoms; cervico-vaginosis caused by sexually transmitted pathogens such as *Chlamydia*, *Neisseria gonorrhoeae,* and *Trichomonas vaginalis*; clinical signs of herpes simplex infection; human papillomavirus or human immunodeficiency virus diseases; both topical or systemic use of probiotic, antibiotic, antifungal, or immunosuppressive drugs during the previous four weeks; any other physiological or pathological condition that could possibly interfere with the results of the study (e.g., genital tract bleeding, chronic illness, cancer, pregnancy, a or breastfeeding). All enrolled patients underwent a clinical and microbiological evaluation at baseline (T0), 15 days after the start of the treatment (T1), and 7 days after the end of the treatment (T2).

The trial was conducted according to current standards of good clinical practice and the Helsinki Declaration (2000) of the World Medical Association. All recruited participants signed a written informed consent before enrolment. The study protocol (n. 16/2022/PO) was approved by the local Ethics Committee (Comitato Etico Catania 1, Azienda Ospedaliero-Universitaria “Policlinico-Vittorio Emanuele” Catania).

### 2.2. Treatment

Recruited patients were subjected to oral administration of the commercial probiotic blend SYNBIO^®^ (SYNBIOTEC Srl, Camerino, Italy), a 1:1 combination of *Lacticaseibacillus rhamnosus* IMC 501^®^ and *Lacticaseibacillus paracasei* IMC 502^®^ (5 × 10^9^ CFU/capsule). Patients were instructed to take one capsule daily for 15 consecutive days.

### 2.3. Clinical Evaluation

Appropriate counseling was offered about the purpose of the study and guaranteed anonymous treatment of personal data in accordance with Italian law. Each participant in the study was informed regarding the procedures that she underwent and signed an informed consent form for data collection. Participants were also advised about the opportunity to freely leave the study at any time should they consider it appropriate. Participation in the study was strictly voluntary, and no remuneration was offered. All eligible participants attending the Gynecological Service were submitted to a preliminary assessment (T0) including the collection of a precise anamnesis and pelvic examination, the evaluation of vaginal symptoms and vaginal pH, and the assessment of Amsel criteria, Nugent score, and lactobacillary Grade, according to Donders criteria.

The anamnesis included age, sexual activity, smoking, weight, height, BMI, chronic illness, diet, and contraceptive use.

Clinical signs and symptoms were evaluated through a severity score on a scale from 0 (absent) to 3 (severe), including the following: leucorrhoea, burning, itching, vulvovaginal erythema/edema subjective vaginal discomfort. 

For every woman, three vaginal swabs were collected. Two vaginal swabs were used to evaluate Amsel criteria and Nugent score: the first one was used for microscopic examination of the fresh smear (detection of clue cells and Gram staining) and the second for the assessment of the Nugent score and whiff-amine test on two different glass slides, The last swab, filled with transport medium, was used for microbiological counts.

The Nugent score was assessed on a 10-point scale, performing a Gram stain followed by optical microscopic observation under oil immersion.

For results analysis, the achievement of at least three of the Amsel criteria was considered as vaginal dysbiosis. A Nugent score of 0–3 was associated with a Lactobacillus-predominant normal vaginal microbiota, a score of 4–6 was interpreted as intermediate, and a score of 7–10 was related to vaginal dysbiosis-like conditions, with the dominance of small Gram-negative and Gram-variable straight and curved rods. Lactobacillary grade (LBG) was evaluated according to Donders classification: LBG I is associated with a normal flora with predominantly lactobacillary morphotypes, and LBG II is characterized by a decreased lactobacillary population mixed with other pathogens, and finally, the severe grade (III) is defined by an abnormal flora that consists of multiple other bacteria and an absence of lactobacilli.

The WHOQOL-BREF questionnaire was administered to assess the quality of life. This instrument can evaluate the global health status of patients through 4 health domains with 24 different domain aspects (i.e., physical, psychological, social relations, and environment). Overall, it includes 26 questions about the previous 2 weeks.

All data were recorded in an appropriate database form, including different sections related to personal data, the patient’s medical history, the intake of any concomitant drugs, symptoms related to vaginal dysbiosis, Amsel score, Nugent score, and microbiological count.

### 2.4. Vaginal Discharge Samples Collection

Vaginal discharge samples were collected from the lateral vaginal wall and the posterior vaginal fornix using sterile cotton-tipped swabs. For every woman, three vaginal swabs were obtained at T0, T1, and T2 sampling times. In detail, two vaginal swabs were used for microscopic examination of the fresh smear (detection of clue cells, Gram staining, evaluation of blastospores and/or pseudohyphae presence) and the whiff-amine test, respectively. In addition, one swab, filled with transport medium (Transystem Amies Medium Clear, Biolife, Milan, Italy), was collected and used for microbiological counts as reported below. Vaginal samples were collected at the Department of General Surgery and Medical Surgical Specialties, Gynecological Clinic, University of Catania (Catania, Italy) and immediately transferred, under refrigerated conditions, to the Laboratory of ProBioEtna Srl (Catania, Italy). Vaginal fluid pH was measured during each visit using pH test strips (McKesson, San Francisco, CA, USA).

### 2.5. Microbiological Analysis

At each sampling time (T0, T1, and T2), vaginal discharge was subjected to culture-dependent analysis. To assure a reliable snapshot of the vaginal microbiota, sterile cotton-tipped swabs filled with transport medium (Transystem Amies Medium Clear, Biolife, Milan, Italy) were used for vaginal discharge sampling. In addition, samples were transferred, under refrigerated conditions, to the Laboratory of Microbiology of the Department of Agriculture, Food and Environment, University of Catania (Catania, Italy) and immediately processed as previously reported by Pino and co-workers [29]. The count of lactobacilli, streptococci, *Gardnerella* spp., *Escherichia coli*, coagulase positive and negative staphylococci, enterococci, and *Candida* spp. was performed using the agar media and conditions reported by Pino et al. [29].

The microbiological count was performed in triplicate, and results were reported as mean log cfu/mL and standard deviation.

### 2.6. Statistical Analysis

Wilcoxon and McNemar tests were performed using the SciPy v1.8.0 Python library to detect significant differences among clinical parameters. One-way ANOVA followed by Tukey’s test, performed using SPSS Version 25.0 (Armonk, NY, USA: IBM Corp.), was applied to detect differences among mean values of the detected microbial groups. Differences were considered statistically significant at a *p*-value < 0.05.

## 3. Results

### 3.1. Demographic, Clinical, and Recruitment Baseline Data

A total of 30 participants meeting inclusion criteria were enrolled. Table 1 shows a summary of both demographics and clinical characteristics of the patients at baseline (T0). Overall, the study population had a mean age of 30.82 years. The majority of patients had sexual activity (86.7%) and were included in the ideal healthy bodyweight range (60.71%). Ten patients were contraceptive users (33.34%). Concerning vulvovaginal signs and symptoms, leucorrhoea and burning occurred most frequently, with percentages of 100% and 96.67%, respectively. At baseline, the percentage of patients who satisfied the Amsel criteria was 83.33% for homogenous vaginal discharge, 73.33% for the presence of clue cells, 56.67% for positivity of the amine test, and 56.67% for the presence of vaginal pH higher than 4.5. In addition, all patients had a Nugent score between 7 and 10, and none showed lactobacillary grade I.

### 3.2. Diagnostic Parameters

The results of the clinical signs or symptoms (leucorrhoea, burning, itching, vulvo-vaginal erythema/edema, and subjective vaginal discomfort) recorded at T0, T1, and T2 sampling times are shown in Figure 1. No adverse events were recorded during the observational period. Compared to baseline (T0), the administration of SYNBIO^®^ for 15 days determined a statistically significant decrease in itching and vulvo-vaginal erythema/edema, whereas leucorrhea was characterized by an increasing trend. Seven days after the end of the treatment (T2), both itching and vulvo-vaginal erythema/edema were characterized by an increasing trend (T1 vs T2 *p* < 0.05), reaching values similar to those observed at baseline (T0 vs. T2 *p* > 0.05). The remaining symptoms did not show a statistically significant change.

According to the clinical results, all the Amsel criteria showed a decreasing trend at T1 sampling time, whereas an opposite tendency was detected at T2 (Figure 2).

Moreover, Figure 3 shows the percentage of patients with a Nugent score higher than 6 at baseline (100%), 15 days after the start of the treatment (17.86%), and 7 days after the end of the treatment (53.57%), revealing a decreasing trend.

Finally, the analysis of the WHOQOL-BREF values revealed the absence of a statistically significant difference in all of the four domains at each sampling time. Consequently, the oral administration of SYNBIO^®^ for the treatment of vaginal dysbiosis did not provide any significant improvement in the quality of life.

### 3.3. Vaginal Microbiota Composition by Microbial Count

Table 2 shows the mean value and standard deviation of the main microbial groups detected at the T0, T1, and T2 sampling times. Overall, at baseline (T0), all recruited patients presented an imbalance of the vaginal microbiota, with a low lactobacilli count and a high cell density of potentially pathogenic bacteria. The oral probiotic administration for 15 days (T1) led to a significant increase in the lactobacilli population, which reached a mean value of viable cells of 5.87 log cfu/mL. Conversely, at the same sampling time, enterococci, staphylococci, *Gardnerella* spp., *Candida* spp., and *E. coli* showed a reduction in cell count. Seven days after the end of the probiotic administration (T2), a quite stable composition of the vaginal microbiota was observed (Table 2).

## 4. Discussion

In this study, we evaluated the effects of the orally administrated probiotic blend SYNBIO^®^ (SYNBIOTEC S.r.l., Camerino, Italy) in women with vaginal dysbiosis. In physiological conditions, normal vaginal flora is dominated by lactobacilli, ensuring a microbiological balance defined as vaginal eubiosis. In fact, the production of lactic acid by the majority of *Lactobacillus* strains has been associated with health benefits due to direct and indirect effects on pathogens and host defense [30]. Conversely, vaginal dysbiosis is a heterogeneous condition associated with the depletion of lactobacilli and a consequent microbiota imbalance and growth of some bacterial species that lead to an increase in susceptibility to infections, hormonal and metabolic disorders [31], as well as negative reproductive outcomes such as infertility [5]. In addition, it has been recently demonstrated that the prophylactic consumption of lactobacillus can prevent preterm birth, providing a potential therapeutic use in pregnancy-associated pathologies [32]. It has also been shown that probiotic treatment from early pregnancy does not modify the vaginal microbiota [33]. Moreover, the use of antimicrobials to counteract BV determined a high recurrence rate of BV, mainly because such treatments are not designed to restore the lactobacilli. Therefore, the rationale for the use of probiotics is based on the need for a restoration of healthy microbiota and the genitourinary regulatory role played by the *Lactobacillus* spp. [34]. Several studies have demonstrated the benefit of oral administration of lactobacilli on vaginal health [34,35], and multiple species of lactobacilli have been evaluated for the treatment of vaginal dysbiosis. In fact, as previously reported, when orally administrated, lactobacilli with probiotic features provide a mechanical barrier against the adhesion of pathogens to the vaginal wall [36]. Our study suggests that the oral application of SYNBIO^®^ once daily for 2 weeks results in a substantial improvement in the vaginal flora of pre-menopausal participants with Nugent scores ≥ 7, as demonstrated by a score reduction by at least two grades with a frequency decrease in patients with a Nugent score higher than 6 from 100% to 17.86% after the two weeks of treatment. This frequency decrease persists even after 7 days of follow up at a percentage value of 53.57%. The probiotic treatment resulted in a significant improvement in the vaginal flora; microbiological data demonstrated that the oral administration of the probiotic SYNBIO^®^ led to a significant increase in the lactobacilli population. Conversely, at the same sampling time, enterococci, staphylococci, *Gardnerella* spp., *Candida* spp., and *E. coli* showed a reduction in cell count. In detail, the vaginal *Candida* spp. count was significantly lower after probiotic treatment, confirming the ability of SYNBIO^®^ to inhibit the growth of *Candida,* as previously demonstrated in vitro [21,22]. The ano-vaginal contamination, well-known for pathogens [37,38], is used in this context also for the passage of probiotics. Another aspect that should not be underestimated is that SYNBIO^®^ is able to survive the gastrointestinal tract passage and to adhere to intestinal epithelium [25,39]. This favors the replenishment through the natural ano-vaginal route for a longer time than local administration, particularly after the end of treatment. Moreover, the two strains used in the blend were also detected in the feces of volunteers for up to one week after the cessation of the treatment [39]. Due to the antimicrobial activity of probiotics in the gut, it could be possible to speculate that the reduced counts of pathogenic microorganisms were the result of fewer pathogens emerging from the rectum and ascending into the vagina. The SYNBIO^®^ oral administration led to a healthier balance of the vaginal microbiota, with a consequent attenuation of some clinical signs and symptoms, such as a significant reduction in itching and vulvovaginal erythema/edema and a significant decrease in all the Amsel criteria at T1. Furthermore, it should be emphasized that the success of a given probiotic treatment to adequately restore the vaginal niche depends primarily on the exact Lactobacillus strains used, but also on the applied dose, formulation, time of administration, and duration of the treatment. Considering that microbiological and clinical data showed an appreciably change during the treatment period, and having ascertained the absence of side effects, it is desirable that a treatment prolonged to 1 month and/or the association of oral treatment with simultaneous intravaginal treatment can provide increased benefits for vaginal health. Nevertheless, we should consider the limitations of this study, including the small number of cases and the absence of a placebo group.

## 5. Conclusions

In conclusion, the present study supports the recent scientific data on the oral use of probiotic strains for restoring the vaginal eubiosis. The use of SYNBIO^®^ has shown improvement in symptoms associated with bacterial vaginosis and also decreases the chances of recurrence by restoring the vaginal microbiota, proving to be a natural, safe, and effective means of fighting vaginal infections. However, a larger two group study is now ongoing and will provide more accurate and reliable results.

## Figures and Tables

**Figure 1 jcm-12-00027-f001:**
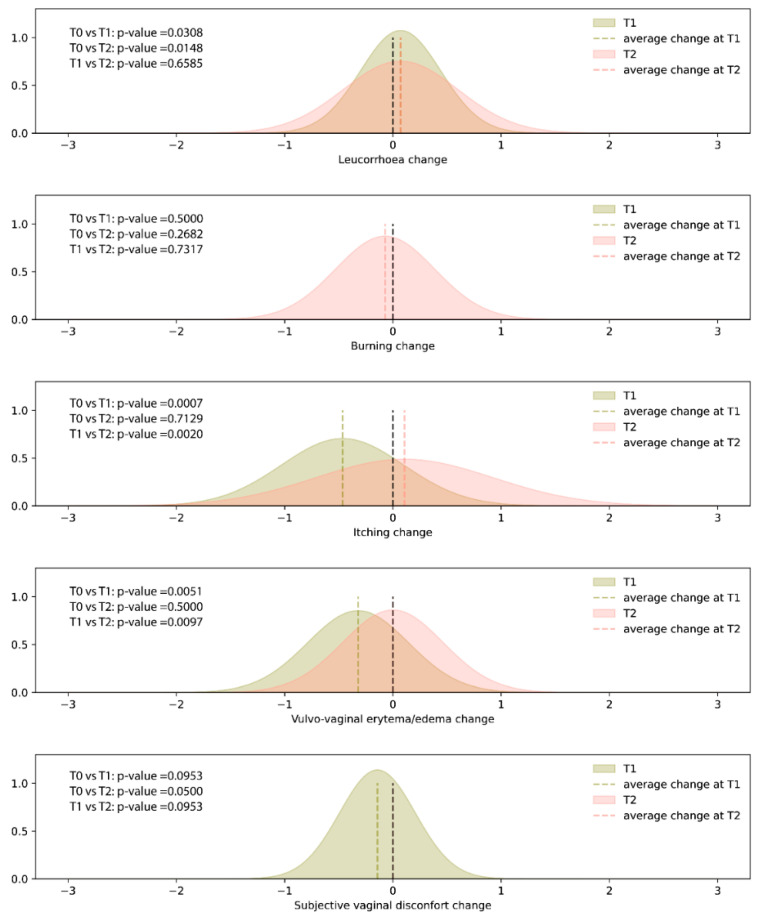
Density plots of the differences in the self-reported signs or symptoms (i.e., leucorrhoea, burning, itching, and subjective vaginal discomfort) scores recorded at T1 (15 days after the start of treatment and T2 (7 days after the end of treatment) sampling times, when compared with T0 (baseline). Dashed lines represent average changes.

**Figure 2 jcm-12-00027-f002:**
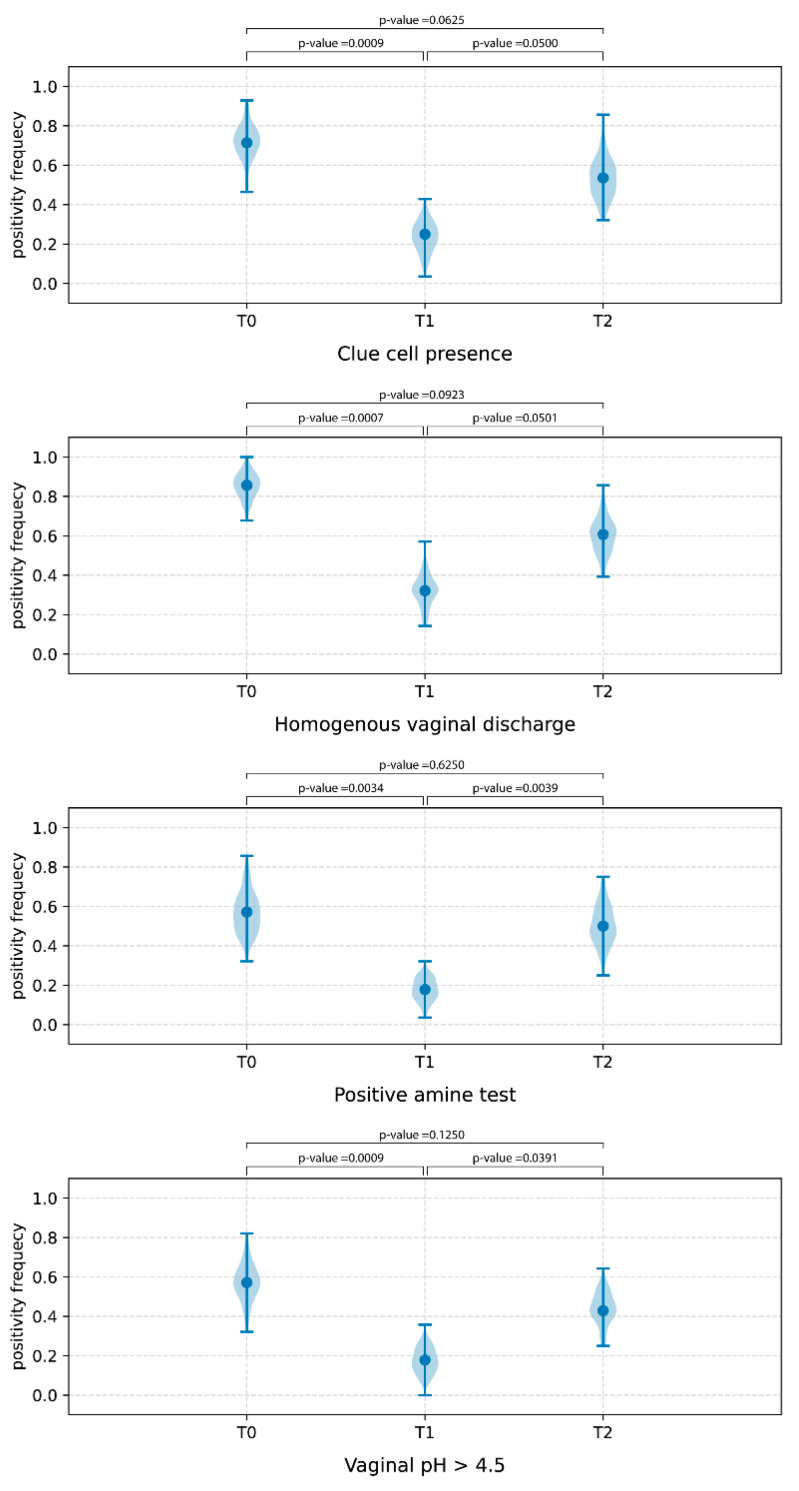
Frequency of positivity of Amsel criteria at T0 (baseline), T1 (15 days after the start of treatment (T1), and T2 (7 days after the end of treatment) sampling times.

**Figure 3 jcm-12-00027-f003:**
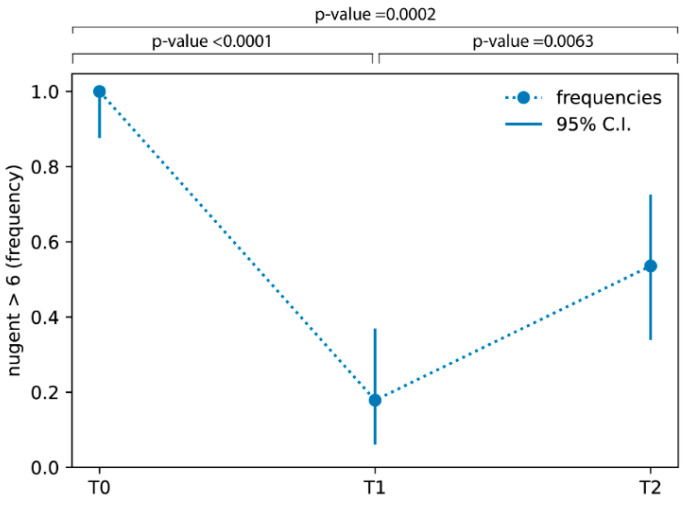
Frequencies of patients with Nugent score higher than 6 at T0 (baseline), T1 (15 days after the start of treatment (T1), and T2 (7 days after the end of treatment) sampling times.

**Table 1 jcm-12-00027-t001:** Baseline demographics and clinical characteristics of the enrolled patients (*n* = 30).

Demographic Characteristics (*n* = 30)			
Age		30.82 ± 6.87	
Sexual activity		26 (86.67%)	
Smoking		14 (46.67%)	
Body mass index (kg/m^2^)		22.94 ± 3.28	
	<18.5	3 (10%)	
	18.5–24.9	17 (56.67%)	
	25–29.9	10 (33.34%)	
	≥30	0 (0%)	
Contraceptive use		10 (33.34%)	
	Oral	5 (16.67%)	
	Barrier	4 (13.34%)	
	Others	1 (3.34%)	
Clinical Characteristics (*n* = 30)			
Vulvovaginal signs and symptoms		Leucorrhoea	30 (100%)
		Burning	29 (96.67%)
		Itching	30 (100%)
		Vulvo-vaginal erythema/edema	30 (100%)
		Subjective vulvar discomfort	30 (100%)
Amsel Criteria		Homogenous vaginal discharge	25 (83.33%)
		Clue cell presence	22 (73.33%)
		Positive amine test	17 (56.67%)
		Vaginal pH > 4.5	17 (56.67%)
Nugent score		0–3	0 (0%)
		4–6	0 (0%)
		7–10	30 (100%)
Lactobacillary grade		I	0 (0%)
		II	6 (20%)
		III	24 (80%)

**Table 2 jcm-12-00027-t002:** Microbial count and ANOVA significance of the vaginal discharge samples collected at baseline (T0), 15 days after the start of treatment (T1), and 7 days after the end of oral SYNBIO^®^ administration (T2). Data are shown as mean and standard deviation.

Microbial Groups	Microbial Count (Log CFU/mL)					
	T0	T1	T2	* *p* T0 vs. T1	* *p* T1 vs. T2	* *p* T0 vs. T2
*Lactobacillus* spp.	3.47 ± 0.77	5.87 ± 0.25	4.13 ± 0.21	0.012	0.082	0.536
*Enterococcus* spp.	4.53 ± 0.39	3.53 ± 0.26	3.63 ± 1.72	0.032	0.936	0.512
*Staphylococcus* spp.	5.03 ± 0.40	3.63 ± 0.47	4.17 ± 0.78	0.035	0.456	0.513
*Streptococcus* spp.	<1	<1	<1			
*Gardnerella* spp.	4.70 ± 0.37	3.43 ± 0.48	3.80 ± 1.51	0.042	0.768	0.456
*Candida* spp.	3.63 ± 0.37	2.57 ± 0.39	3.23 ± 0.12	0.043	0.086	0.213
*Escherichia coli*	4.37 ± 0.21	2.77 ± 0.82	2.87 ± 0.37	0.045	0.881	0.007

* Statistical significance *p*  <  0.05.

## Data Availability

Not applicable.

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
