# Peer review of "Oral Intake of the Commercial Probiotic Blend Synbio® for the Management of Vaginal Dysbiosis"

_jcm, 2022, doi:10.3390/jcm12010027_

Round 1

Reviewer 1 Report

The work by Pino and co-authors entitled “Oral intake of the commercial probiotic blend Synbio® for the 2 management of vaginal dysbiosis “is an open-label study that aims to investigate the capacity of a commercial probiotic formulation to influence vaginal health.

Though the results are very interesting and seem to clearly demonstrate a beneficial effect of probiotic consumption on vaginal health, a placebo group is missing. This is a critical issue, and the author should either include a placebo group in their analysis or clearly state that this is an “uncontrolled” open-label study. The first option would be more appropriate.

The authors barely mention the already-known positive effect of probiotic consumption on vaginal health and its potential therapeutic use in pregnancy-associated pathologies like preterm birth (PTB). Indeed, it has been recently demonstrated that prophylactic consumption of lactobacillus can prevent PTB in a mouse model of the disease (DOI: 10.1530/REP-20-0642). Moreover, it has also been shown that probiotic treatment from early pregnancy does not modify the vaginal microbiota (reviewed in DOI: 10.1016/j.jri.2022.103493).

Hence, all this information should be discussed and accordingly cited in this work.

Author Response

The work by Pino and co-authors entitled “Oral intake of the commercial probiotic blend Synbio® for the 2 management of vaginal dysbiosis “is an open-label study that aims to investigate the capacity of a commercial probiotic formulation to influence vaginal health.

 Though the results are very interesting and seem to clearly demonstrate a beneficial effect of probiotic consumption on vaginal health, a placebo group is missing. This is a critical issue, and the author should either include a placebo group in their analysis or clearly state that this is an “uncontrolled” open-label study. The first option would be more appropriate.

Thanks for the insightful comment. We agree with the critical issue concerning the missing placebo group. The present study was conceived as a preliminary study aimed to evaluate the ability of the SYNBIO® probiotic combination, taken as an oral formulation, to contribute to vaginal health. A placebo-controlled study with a higher number of enrolled patients is now ongoing. As suggested, we clearly stated in the text that the present study was an “uncontrolled” open-label study. 

The authors barely mention the already-known positive effect of probiotic consumption on vaginal health and its potential therapeutic use in pregnancy-associated pathologies like preterm birth (PTB). Indeed, it has been recently demonstrated that prophylactic consumption of lactobacillus can prevent PTB in a mouse model of the disease (DOI: 10.1530/REP-20-0642). Moreover, it has also been shown that probiotic treatment from early pregnancy does not modify the vaginal microbiota (reviewed in DOI: 10.1016/j.jri.2022.103493).

Thank you for the suggestion. We revised the discussion, highlighting the positive effect of probiotic consumption in pregnancy-associated pathologies like preterm birth (PTB) (DOI: 10.1530/REP-20-0642). Moreover, we emphasized that probiotic treatment from early pregnancy does not modify the vaginal microbiota (reviewed in DOI: 10.1016/j.jri.2022.103493). We cited the references as suggested.

Reviewer 2 Report

This is a very well-done preliminary study.  Well written scientific rationale for the probiotic intervention. 

*It would be clearer to use Bacterial Vaginosis in the title of your article.  BV is certainly a form of vaginal dysbiosis.  Your inclusion criteria were women with BV.   The study outcomes you used are those used for BV (except “leucorrhoea”-see below**).  BV is a more clinically relevant term. 

**”Leucorrhoea” is generally not considered part of the clinical or diagnostic constellation of BV symptoms.  Can you explain why you used that outcome and what you mean by that term? In the USA we use the term "Leukorrhea" for the increased vaginal discharge that is a normal physiologic change of pregnancy.  How do you use that term in Italy? Do you mean increased discharge associated with BV?

*The inclusion of the background on VVC in the background detracts from the focus on BV. Therefore you may consider deleting the following:

[Concerning VVC, a common vaginal mucosa infection 58 caused by Candida species, vulvar pruritus and burning as well as vaginal dyspareunia 59 and dysuria are the main clinical symptoms. It is estimated that more than 75% of women 60 experienced at least once Candida infection in their lifetime and that recurrent VVC is fre-61 quent [12,13]. That’s why, preserving a balance of vaginal microbial composition is essen-62 tial to guarantee an adequate host-microbial interaction that ensure a healthy vaginal mi-63 crobiota.]

*What was the rationale for 15 days of SYNBIO®? 

*This reviewer would recommend using "participants" versus "women" to refer to the 30 people you studied.

* 26 (86.67%) of the participants were sexually active, but only 10 (33.34%) were using contraception.  Do you have more information about this issue? For example; Does that mean that participants are trying to conceive? ...or Are having sex with women? ... or Are having oral or anal intercourse with a male partner? ...or some combination of the above? 

*Your micro biome analysis was interesting and well done. 

Figure 1 contains excellent graphs of BV symptom changes.

Figure 2 Can you explain a bit more and why you think the components of the Amstel’s criteria reverted back to baseline levels at T3 after showing a significant change at T2? In other words, do you think 15 days of the probiotic was too brief to sustain the change?

I hope you do a larger two group study. 

Author Response

This is a very well-done preliminary study.  Well written scientific rationale for the probiotic intervention.

*It would be clearer to use Bacterial Vaginosis in the title of your article.  BV is certainly a form of vaginal dysbiosis.  Your inclusion criteria were women with BV.   The study outcomes you used are those used for BV (except “leucorrhoea”-see below**).  BV is a more clinically relevant term.

Many thanks. We are pleased with this encouraging comment. In our study, we used the term vaginal dysbiosis to include the high heterogeneity of microbiological and clinical findings observed in gynecological practice. In detail, it is essential to highlight that the term vaginal dysbiosis includes a wide-ranging panel of vaginal conditions that have, as their common denominator, the decreasing number of lactobacilli and the overgrowth of pathogens. Indeed, although strict clinical and microbiological features allow categorizing vaginal conditions (bacterial vaginosis, aerobic vaginitis or vulvovaginal candidiasis) in everyday practice, different species of pathogens, both aerobic, anaerobic and fungi, usually coexist. This heterogeneity leads to many clinical traits that can hardly be adapted to rigorous theoretical definitions and criteria. Considering the defined enrolled criteria of our study (women with signs and symptoms of vaginal impairment confirmed by clinical and microbiological evaluation), we consider the term vaginal dysbiosis as more appropriate in the title.

**”Leucorrhoea” is generally not considered part of the clinical or diagnostic constellation of BV symptoms.  Can you explain why you used that outcome and what you mean by that term? In the USA we use the term "Leukorrhea" for the increased vaginal discharge that is a normal physiologic change of pregnancy.  How do you use that term in Italy? Do you mean increased discharge associated with BV?

Thank you for your insightful comment. We used the term Leucorrhea as a patient self-reported symptom of an abnormal increased vaginal discharge. Indeed, even if “Leucorrhea” is not specifically related with bacterial vaginosis, it is well known the link between changes in vaginal discharge and imbalance of the vaginal microbiota (Böcher S, Helmig RB, Arpi M, Bjerrum L. Ugeskr Laeger. 2018;180(3):V03170229) (Romero Herrero D, Andreu Domingo A. Vaginosis bacteriana [Bacterial vaginosis]. Enferm Infecc Microbiol Clin. 2016;34 Suppl 3:14-18. doi:10.1016/S0213-005X(16)30214-2).  For instance, a such imbalance could be related to a vulvovaginal candidiasis.

Therefore, we used the “Leucorrhea” as an additional parameter to achieve a more accurate self-reported evaluation of vaginal dysbiosis.  

*The inclusion of the background on VVC in the background detracts from the focus on BV. Therefore you may consider deleting the following:

[Concerning VVC, a common vaginal mucosa infection 58 caused by Candida species, vulvar pruritus and burning as well as vaginal dyspareunia 59 and dysuria are the main clinical symptoms. It is estimated that more than 75% of women 60 experienced at least once Candida infection in their lifetime and that recurrent VVC is fre-61 quent [12,13]. That’s why, preserving a balance of vaginal microbial composition is essen-62 tial to guarantee an adequate host-microbial interaction that ensure a healthy vaginal mi-63 crobiota.]

Thanks for the remark. In our study, we investigated the management of women with vaginal dysbiosis, including participants with mild and moderate vaginal symptoms. Therefore, we included even the cases of mixed vaginitis with symptoms related to the presence of both Candida spp. and other bacterial pathogens. Indeed, Candida species' interaction could influence symptoms' development and severity.

Nevertheless, we considered as exclusion criteria the presence of severe vulvovaginal symptoms related to Candida infections and the need for appropriate antifungal therapy.

Hence we prefer not modify the sentence you suggested

 *What was the rationale for 15 days of SYNBIO®?

Thanks for the insightful comment. The rationale for the administration of SYNBIO® for 15 days was based on an extensive review of the literature focused on oral lactobacilli to treat vaginal dysbiosis. As recently confirmed by Chee and co-workers (2020), although trials report large variations in terms of treatment duration, several studies have shown that the oral administration of lactobacilli for 10-15 days was able to balance the vaginal microbiota (De Alberti et al., 2015; Mezzasalma et al., 2017; Pino et al., 2021). Based on our unpublished preliminary data, we hypothesized that treatment of 15 days could be effective for treating vaginal dysbiosis.

*This reviewer would recommend using "participants" versus "women" to refer to the 30 people you studied.

Thanks for the suggestion. We referred to “participants” rather than “women” throughout the text. 

* 26 (86.67%) of the participants were sexually active, but only 10 (33.34%) were using contraception.  Do you have more information about this issue? For example; Does that mean that participants are trying to conceive? ...or Are having sex with women? ... or Are having oral or anal intercourse with a male partner? ...or some combination of the above?

Thanks for the question. We showed the percentage of sexually active women using contraception in consideration of the known link between vaginal microbiota and contraceptive methods. (Bastianelli C, Farris M, Bianchi P, Benagiano G. The effect of different contraceptive methods on the vaginal microbiome. Expert Rev Clin Pharmacol. 2021;14(7):821-836. doi:10.1080/17512433.2021.1917373, Balle C, Konstantinos IN, Jaumdally SZ, et al. Hormonal contraception alters vaginal microbiota and cytokines in South African adolescents in a randomized trial. Nat Commun. 2020;11(1):5578. Published 2020 Nov 4. doi:10.1038/s41467-020-19382-9)

Nevertheless, we did not investigate why sexually active women used or did not use a contraceptive because it was not an established outcome in the study design.

*Your micro biome analysis was interesting and well done.

Figure 1 contains excellent graphs of BV symptom changes.

Figure 2 Can you explain a bit more and why you think the components of the Amstel’s criteria reverted back to baseline levels at T3 after showing a significant change at T2? In other words, do you think 15 days of the probiotic was too brief to sustain the change?

I hope you do a larger two group study.

Thanks for your comment. As correctly underlined, we noticed a back revetement trend in Amsel’s criteria results at the T3 follow-up. The clinical finding can be influenced by many variables such as diet, properties, and interactions of lactobacilli. Our data suggest that 15 days oral administration of the SYNBIO® probiotic combination is not enough to preserve the vaginal eubiosis. Further studies with more patients and related clinical outcomes is now ongoing to evaluate the persistence of probiotic combination, mentioned above, in the vaginal ecosystem.

Round 2

Reviewer 1 Report

The authors have fully addressed the issues raised by this reviewer and accordingly introduced the suggested modifications